# Characteristics and Mechanism of Local Scour Reduction around Pier Using Permeable Sacrificial Pile in Clear Water

**Hongliang Qi [1],\*, Guishan Chen [1] , Wen Zou [2], Tiangang Yuan [1], Weiping Tian [1] and Jiachun Li [1]**

1   Key Laboratory for Special Area Highway Engineering of Ministry of Education, School of Highway, Chang'an University, Xi'an 710064, China
2   Hexagon Software Technology Co., Ltd., Qingdao 266000, China
\*   Correspondence: qihongl@chd.edu.cn; Tel.: +86-29-8233-4443

**Abstract:** To improve the local scour protection of the pier using solid sacrificial piles, a kind of permeable sacrificial pile filled with stones is put forward in this study. Four influencing factors, including the size of the filling gravel of the permeable sacrificial pile, the distance between the pile and pier, the diameter ratio D1/D (D is the diameter of the pier and D1 is the diameter of the pile.), and the submergence rate of the pile, are studied in clear water condition with indoor tests and numerical simulations. The test results show that it has the best reduction effect on the local scour of the pier when the filled gravel size is 0.2–0.25 D, the distance between the pile and pier is 3.0 D, the diameter ratio is 1.0, and the submergence rate is 0.8. Results of the numerical simulation show that the permeable sacrificial pile has a significant weakening effect on the downflow in front of the pile and the flow velocity around the pile. In the same conditions, the reduction effects of the permeable sacrificial pile and the solid sacrificial pile on the pier are similar, but the local scour around the permeable sacrificial pile is less than that around the solid one. The pressure difference outside and inside the permeable pile causes the water to flow in the direction perpendicular to the isobars. This significantly impacts the flow velocity, the vortex system, and the shear stress around the permeable sacrificial pile and pier, which leads to a huge decrease in the local scour depths. The maximum shear stress on both sides of the permeable sacrificial pile and the pier is about 1/2 and 1/4 of those around the solid sacrificial pile respectively. And the area with lower velocity and shear stress behind the permeable sacrificial pile is larger than the area behind the solid ones.

**Keywords:** permeable sacrificial pile; pier; local scour; model test; numerical simulation

## 1. Introduction

Local scour is one of the main causes of bridge collapses [1–4]. Many studies on the mechanism of local scour, calculation of local scour depth, and measurements of local scour reduction have been carried out all over the world.

Gao and Wang [5,6] analyzed significant impacts on the flow velocity, the vortex system, and the shear stress around the permeable sacrificial pile and pier. The abutment and the main factors were put forward. Tian [7–10] studied the flow patterns and scour law of spur dike groups in straight and curved rivers and summarized the various characteristics and layout requirements of spur dike scour depth in bends. Their research enriches the scour protection types and layout methods of subgrade along curved and concave banks of the river. Raeisi et al. [11] studied the scale effect of cylindrical piers on local scour with model tests. The research shows that for piers of diameter under 30 mm, the relative scour depth is purely a function of the Froude number. Nazila et al. [12] studied the effects of different angles, flow intensities, flow depths, and pier sizes on local scour. The results show that the scour depth increases with the increase in pier width. The scour depth of the inclined pier is less than that of the vertical pier. The maximum scour depth decreases with the increases in pier inclination. Wang et al. [13] studied the development of local scour

around cylindrical piers in different flow velocities and water depths. The research shows that the scour depth and scope of scour pit develop rapidly at early stage, and slow down afterwards until reaches to an equilibrium stage.

With the development of computer science and numerical methods, Computational Fluid Dynamics applications are widely used, such as Flow-3D, ANSYS Fluent, etc. Flow-3D is a general CFD software released by the American Flow Science Company in 1985. It is an efficient fluid computing simulation tool. ANSYS Fluent is a widely used CFD application all over the world. Its applicability and accuracy have been proved by many scholars. Qi et al. [14] studied the effects of longitudinal pier span on the characteristics of excess shear stress near the riverbed with FLOW-3D. Duan et al. [15] studied the local scour around double-row cylindrical piers on non-uniform sandy riverbeds using FLOW-3D. Their research shows that the scour pit runs through the whole area around the pier. The depth and scale of the local scour pits around downstream piers are less than those of the upstream piers because the upstream piers protect the downstream ones. Wang et al. [16] simulated the process of local scour around piers by using a self-defined function and technology for dynamic grid updating in FLUENT. The maximum depth of the local scours, the shape of the scour pit, and the flow field structure agreement with the experimental results. Guo et al. [17] simulated the scour process of the bridge pier using the VOF technology of FLOW-3D. The velocity and scour distribution near the pier are in good agreement with classical scour tests.

Sacrificial piles set in front of piers is an effective way to reduce the local scour of piers. Jorge et al. [18] analyzed the influence of the diameter and the distance of the solid sacrificial piles on reducing the local scour depth around the piers. The results show that, the longer the distance between the solid sacrificial pile and the pier, the higher the local scour reduction percentage. Jiang et al. [19] studied the effects of diameter ratio, layout distance, and submergence rate of solid sacrificial piles on pier column scour protection under constant flow conditions. The results show that, when diameter ratio increases, the local scour of the pier column slows down. With the decrease of submergence rate, the protection effect decreases gradually. If the solid sacrificial pile is too close or too far from the pile, the protection effect is not obvious. Melville et al. [20] studied the protective effects of solid sacrificial piles on local scour around piers in different layout forms. The research shows that the protective effect is better when the solid sacrificial piles are arranged in a triangle in clear water conditions. In most scenarios, the reduction effect of the submerged pile is more significant than unsubmerged pile. Chen et al. [21] studied the protective effect of the solid sacrificial piles in the forms of single and group through model tests and numerical simulations. The research shows that the solid sacrificial piles can effectively reduce the local scour depth of the pier in whichever form. Zhang et al. [22] studied the protective effect of a solid sacrificial pile on the local scour of piers through numerical simulations. The results show that the solid sacrificial pile can reduce the flow velocity, turbulence intensity, and velocity near the riverbed around the pier. Haque et al. [23] studied the reduction effect of the solid sacrificial pile on the local scour of the pier. The results show that the local scour depth around the pier can be reduced up to 61%.

Above all, scholars all over the world have explored the mechanism of local scour, protective measures, and the calculation methods of scour depth. The solid sacrificial pile can effectively reduce the local scour depth of the pier. However, the local scour of the solid sacrificial pile is also very serious, which is a great threat to the safety of the sacrificial pile. At present, little research on the reduction of local scour around sacrificial piles has been reported.

To reduce the local scour of the sacrificial pile, a permeable sacrificial pile filled with stone is proposed in this paper. The main influencing factors of local scour, including the size of the filling gravel which represents permeability of the sacrificial pile, the layout distance between the pile and the pier, the diameter ratio, and the submergence rate of the permeable sacrificial pile, are studied. The mechanism of local scour reduction is revealed

based on indoor experiments and numerical simulations. The enhanced sacrificial piles and research results can provide a reference for optimizing local scour protection of bridge piers.

## 2. Experiment Setup and Procedures

### 2.1. Experiment Equipments

The experiments were conducted in a 20.0 m long, 1.6 m wide, and 0.7 m deep open channel in the hydraulic laboratory of Chang'an University, Shaanxi, China. The water depth was 0.2 m, shown in Figure 1. To make sure the water flows into the test section smoothly, a 7.0 m long fixed bed section was set at the inlet of the flume with 5 flow straighter. The length of the test section was 7.0 m and the pier was located in the middle of the test section. The pier was a cylindrical pipe made of PVC with a height of 50.0 cm and a diameter of 8.0 cm. The permeable sacrificial pile was made of stainless-steel mesh and filled with stones. The diameter of the permeable pile as D1, the distance between the center of the permeable pile and the pier as L, and the height of the permeable pile above the sediment surface as h, are shown in Figure 1b. The test section was covered with 0.15 m thick uniform sediment. The median particle size of the sediment $d_{50} = 0.89$ mm, the non-uniformity coefficient Cu = 7.16, and the curvature coefficient Cc = 0.95. The sediment is well-graded and continuous and is qualified for the test. The grading characteristics of the sediment are shown in Figure 2.

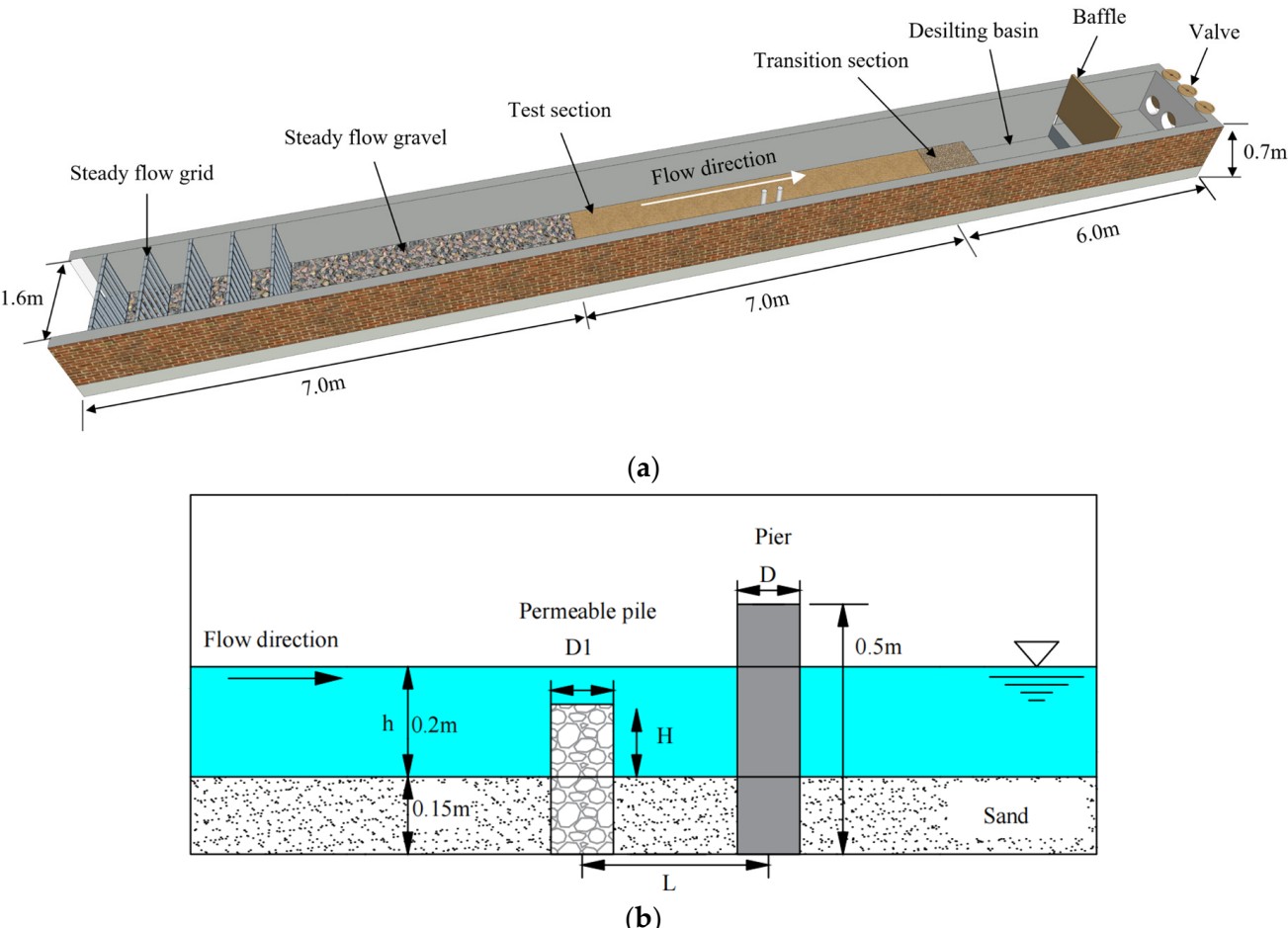

**Figure 1.** Layout of the test system. (**a**) Top view of the test system. (**b**) Side view of the test section.

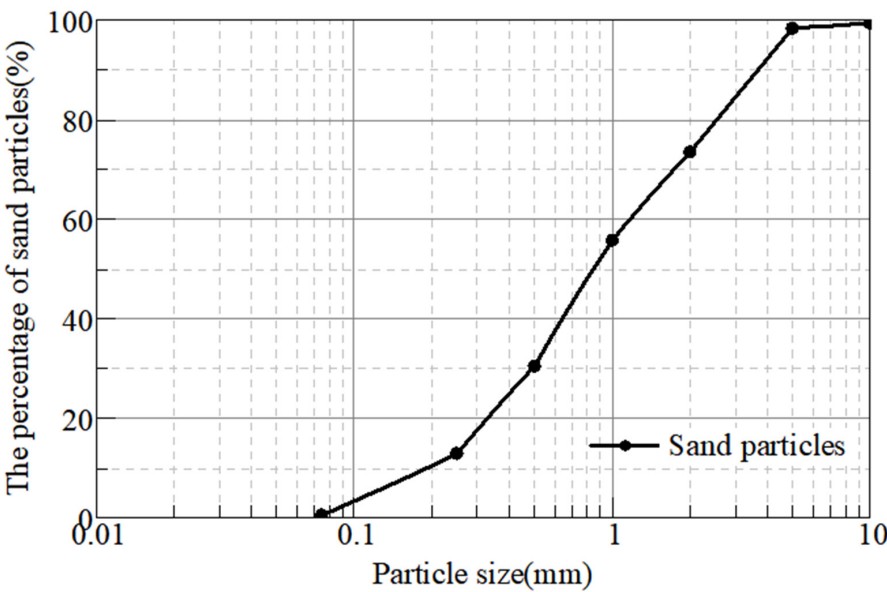

**Figure 2.** Grading curve of sediment.

### 2.2. Design of the Experiments

In this study, the effects of four factors, including the average diameter (*d*) of the stones filled in the permeable pile, the distance (*L*) between the permeable pile and pier, the diameter ratio ($D_1/D$), and the submergence rate (H/h), on the local scour reduction characteristics of the permeable sacrificial pile were studied in the condition of uniform flow. There were 6 groups of experiments in this study, named A, B, C, D, E, and F, shown in Table 1. Group A was designed for local scour around the pier using an unsubmerged solid sacrificial pile. The diameter of the pile was the same as that of the pier, and *L* was 4.0 D. Group B studied the influence of gravel particle size of the permeable pile on local scour reduction. There were five levels: 8–12 mm (0.1–0.15 D), 12–16 mm (0.15–0.2 D), 16–20 mm (0.2–0.25 D), 20–24 mm (0.25–0.3 D), and 24–28 mm (0.3–0.35 D). The ratio of the maximum length to the minimum thickness of the filler gravel was less than 2, and the porosity for each gravel level was 0.445 mm, 0.457 mm, 0.491 mm, 0.493 mm, and 0.497 mm, respectively. Group C focused on the influence of L, and six levels were taken: 2.0 D, 3.0 D, 4.0 D, 5.0 D, 6.0 D, and 7.0 D. Group D studied the influence of the diameter ratio (D1/D) on the local scour reduction, and five levels were selected: 0.6, 0.8, 1.0, 1.2, and 1.5. Group E studied the influence of submergence rate (H/h) on local scour reduction at five levels: 0.2, 0.4, 0.6, 0.8, and 1.0. The layout of the experimental model is shown in Figure 3. The permeable sacrificial pile was a cylinder-shaped steel grid and filled with gravel.

**Table 1.** Factors and levels of the experiments.

| Group | Experimental Factors | Levels | Filling Particle Size | Layout Distance | Diameter Ratio | Submergence Rate |
|---|---|---|---|---|---|---|
| A | - | Solid sacrificial pile | - | 4.0 D | 1.0 | 1.0 |
| B | Particle size (*d*) | 0.1–0.15 D, 0.15–0.2 D, 0.2–0.25 D, 0.25–0.3 D, 0.3–0.35 D | - | 4.0 D | 1.0 | 1.0 |
| C | Distance (*L*) | 2.0 D, 3.0 D, 4.0 D, 5.0 D, 6.0 D, 7.0 D | Optimum particle size | - | 1.0 | 1.0 |
| D | Diameter ratio ($D_1/D$) | 0.6, 0.8, 1.0, 1.2, 1.5 | | Optimum distance | - | 1.0 |
| E | Submergence rate (H/h) | 0.2, 0.4, 0.6, 0.8, 1.0 | | | 1.0 | - |

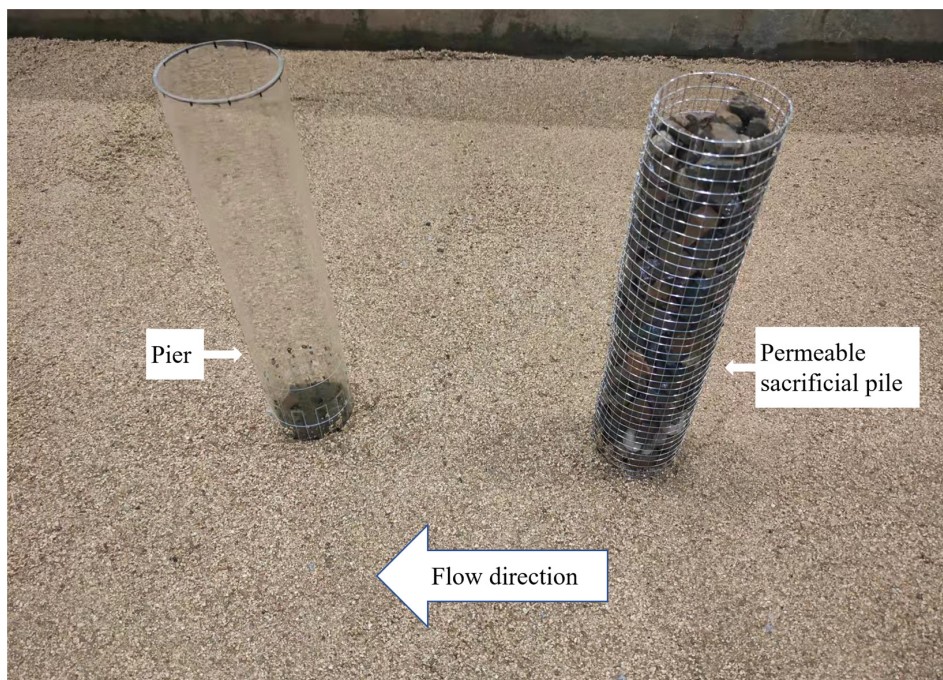

**Figure 3.** Model of the pier local scour reduction with the permeable sacrificial pile.

The water flow in the test was supplied by a pump with a maximum water supply of 200 L/S. The flow rate in this study was 100 L/S. The starting velocity of sediment in this study was 0.349 m/s, which was calculated according to Zhang Ruijin's formula. The vertical flow velocity at 1.5 m in front of the pier was measured with a flow velocity instrument. The actual flow velocity was 0.306 m/s calculated by following the five-point method. It was smaller than the starting velocity. Therefore, the experiments were carried out in clear water.

The development of the local scour around the pier was recorded by an underwater camera. The topography of local scour around the model was measured by a laser rangefinder with a grid size of 1.0 cm × 1.0 cm.

## 3. Experiment Results and Analysis

### 3.1. Local Scour around a Single Pier

For later analyses and comparisons, local scour tests around a single pier without protection were carried out at first. It was tested three times in this scenario. It turned out that the shapes, depths, and scopes of the scour holes were very close. Therefore, one test was carried out in each subsequent working conditions. The flow rate and water depth were both observed in the process to ensure the conditions of each test were the same.

At the beginning, a downward flow was formed in front of the pier. When the downward flow reached the sediment surface, a clockwise rotating vortex was formed in front of the pier, which brushed the sediment away and carried them downstream. The local scour depth and range around the pier increased rapidly. With the expansion of the local scour scope, the flow velocity around the pier decreased gradually which caused the local scour depth and the scope to change gradually. Finally, the local scour reached the scour equilibrium. The above phenomena and processes are consistent with existing research results.

Figure 4 shows the local scour characteristics around a single pier without protection, with a local scour depth of 7.1 cm. The relationship between the local scour depth in front of the pier and time is shown in Figure 5. The development of local scour depth in front of the pier is consistent with the existing research. The process reached to the scour equilibrium within 600 s. So, the time of the model test in this study was set to 60 min to make sure later experiments can reach to the scour equilibrium.

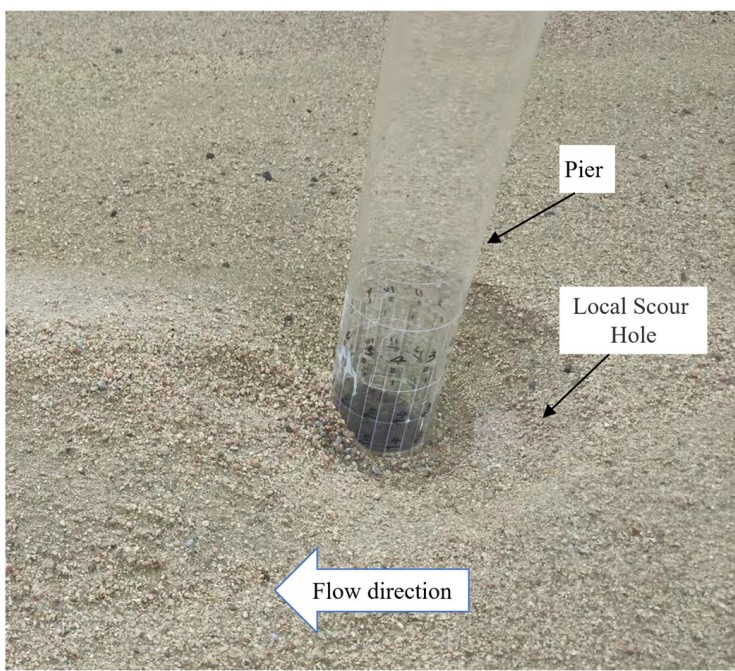

**Figure 4.** Local scour around a single pier.

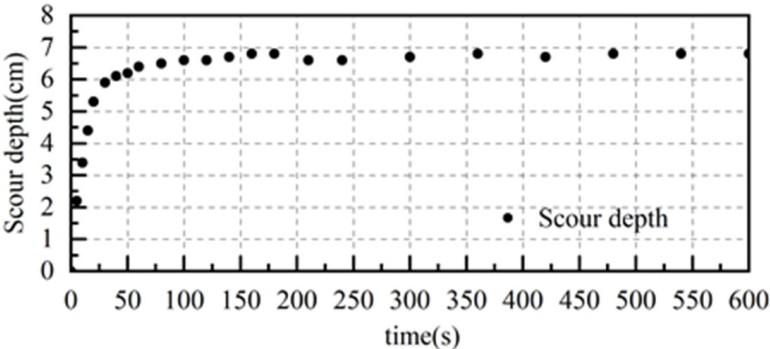

**Figure 5.** Development of local scour depth around a single pier.

*3.2. Influence of Gravel Size on the Protective Effect of Permeable Sacrificial Pile*

The local scour characteristics around the pier and permeable sacrificial pile filled with different gravel sizes are shown in Figure 6. From Figure 6A, it can be concluded that the solid sacrificial pile effectively reduced local scour around the pier, but the local scour around itself was very serious. The local scour depth around the permeable sacrificial pile is smaller than that around the solid one, which means that the introduction of permeability can reduce the local scour depth of the sacrificial pile. Whilst the protection effect of the permeable sacrificial pile on local scour around the pier is close to that of the solid one compared to experiment A. The regions of local scour around the pier in experiment group B were larger, and the scour depths behind the permeable sacrificial pile were smaller. Deposition even happened between the permeable sacrificial pile and the pier. Therefore, the permeable sacrificial pile affects effectively not only the local scour, but also the regional scour around. Due to the barrier of the pier, the cross-sectional area is reduced, which causes a rapid increase in the flow velocity around the front of the pier. The increase in the flow velocity causes local scouring, the sediment around the pier is carried to the downstream area of the pier. Due to the vortex shedding of the pier, the flow velocity is much lower in a certain range in the downstream area of the pier. The sediment from upstream was deposited in the lower-velocity area gradually.

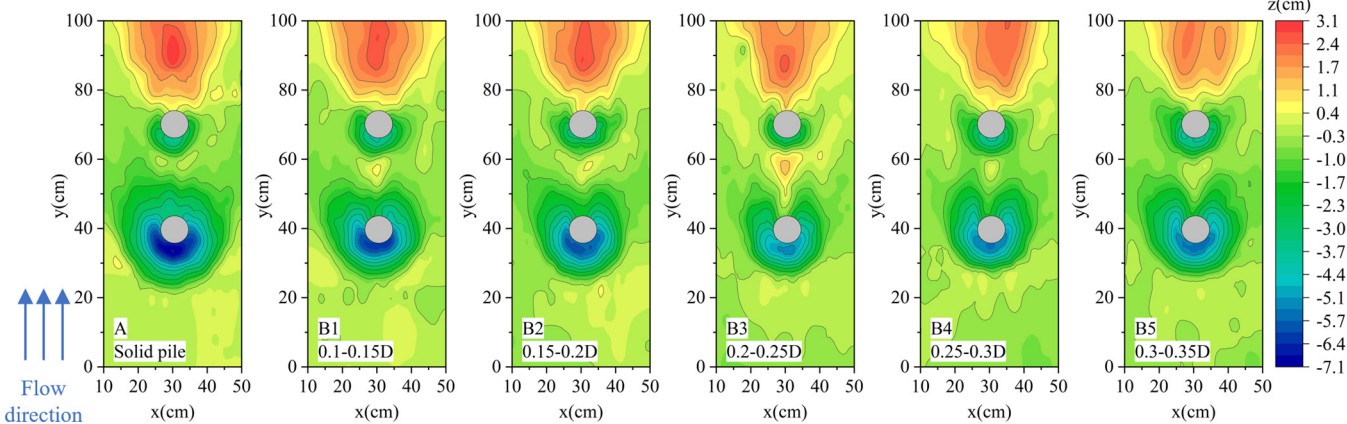

**Figure 6.** Local scour around the pier protected by permeable sacrificial pile with different particle sizes.

Figure 7 shows the local scour depths around the pier and permeable sacrificial pile filled with different gravel sizes. With the increase of the gravel size, the local scour depth of the permeable sacrificial pile tends to decrease and then increase. The local scour depth of the permeable sacrificial pile reached the minimum when the gravel size is 0.2–0.25 D. The value around the pier is about 85.71% of that in case A. Compared with case A, the local scour depth around the permeable sacrificial pile is reduced at 11.27%, 16.90%, 28.17%, 22.54%, and 18.31%, respectively. Above all, the permeable pile can provide close protection as that of solid pile, but also reduce the local scour depth of itself in the meantime.

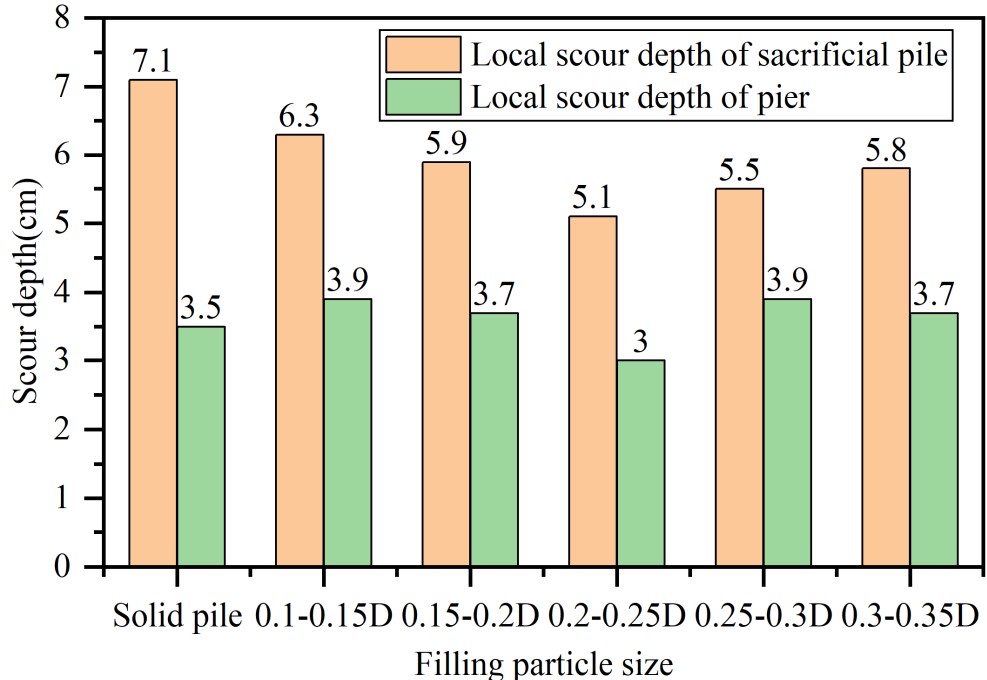

**Figure 7.** Local scour depth around the pier and permeable sacrificial pile with different gravel sizes.

The variation of porosity is the main reason for the trend of the local scour depth around the permeable sacrificial pile which decreases and then increases with the increase of the particle size. The porosity of the permeable sacrificial pile filled with different gravel sizes was 0.445, 0.457, 0.491, 0.493, and 0.497, respectively. The pore space inside the permeable sacrificial pile increases with the increase in gravel size. When the gravel size is small, the porosity of the permeable sacrificial pile is small, as well as the permeability. However, the compactness is large, which means that there is more downward flow in front of the pier, so serious local scour around the permeable sacrificial pile is caused. With

the increase in porosity, the permeability increases as well, and more water flows through the permeable pile, which leads to a decrease in the downward flow in front of the pier. The local scour depth around the permeable sacrificial pile decreases. If the gravel size becomes much larger, although the porosity of the permeable sacrificial pile increases, the flow path inside the permeable pile is reduced. Meanwhile, larger particle size means a larger water blocking, which increases the downward flow in front of the pile and leads to a deeper local scour around the permeable sacrificial pile.

### 3.3. Influence of Layout Distance on the Protect Effect of Permeable Sacrificial Pile

Figure 8 shows the local scour topography around the pier and permeable sacrificial pile filled with gravel size of 0.2–0.25 D at different layout distances. The layout distance from C1 to C6 was 2.0 D, 3.0 D, 4.0 D, 5.0 D, 6.0 D and 7.0 D, respectively. When the layout distance is less than 3.0 D, the scour holes around the permeable sacrificial pile and pier joined together, and there is barely any sediment siltation behind the permeable sacrificial pile. When the distance is greater than 3.0 D, the local scour holes around the pile and pier depart from each other, and the local scour depth around the piers becomes greater. However, if the layout distance keeps on increasing, the local scour depth and range of the permeable sacrificial pile would finally remain stable.

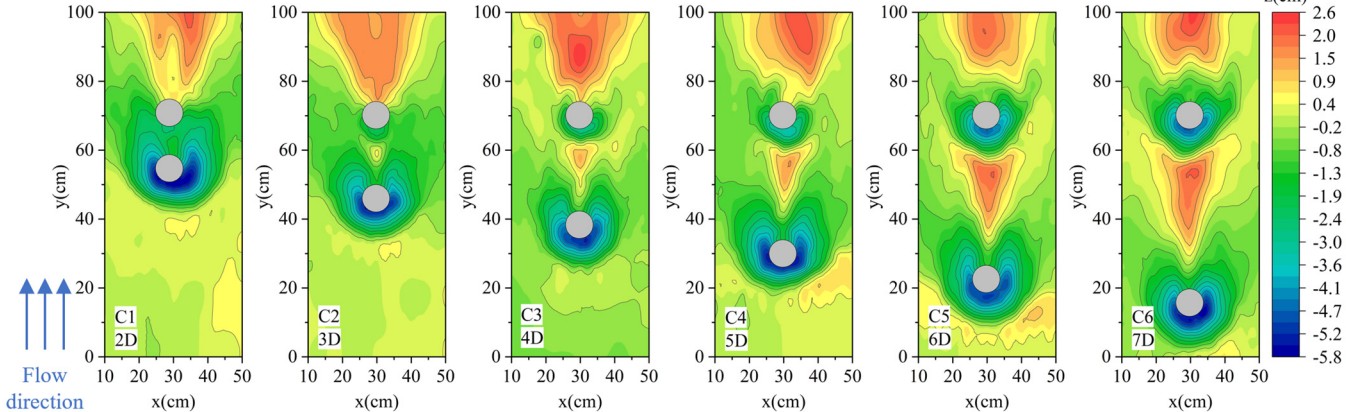

**Figure 8.** Topographic maps of scour around pier protected by permeable sacrificial pile with different distances.

Figure 9 shows the local scour depths of the permeable sacrificial pile and the pier in different layout distances. When the distance between the pile and the pier increases, the local scour depth of the pier decreases, then increases, and finally reach stability. When the layout distance is 3.0 D, the local scour depth of the pier is the smallest, and the protective effect on the pier is the best. When the distance is longer than 5.0 D, the local scour depth becomes stable and does not increase significantly. In an overall view of the local scour depth for both the permeable sacrificial pile and the pier, the protection effect is the best at the layout distance of 4.0 D.

According to the local scour characteristics around the cylinder, the sediment near the cylinder is brushed by the horseshoe vortex and brought downstream, and the sand is deposited in a certain area behind the cylinder. If a downstream pier is located in the siltation area of the upstream cylinder, its local scour depth will be reduced. Otherwise, the local scour becomes more serious. If the downstream pier is away from the upstream cylinder, the local scour depth will gradually increase until stable, like two independent piers. Then the local scour depth of the downstream pier reaches the maximum.

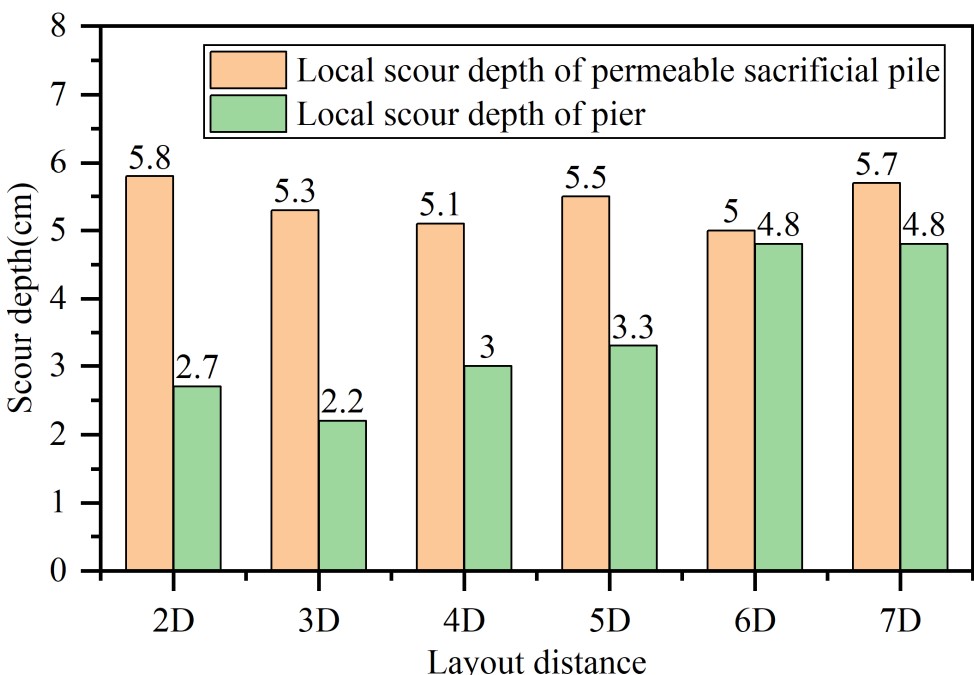

**Figure 9.** Local scour depths around the pier and permeable sacrificial pile with different distances.

*3.4. Influence of Diameter Ratio on the Protect Effect of Permeable Sacrificial Pile*

Figure 10 shows the topography of local scour around the permeable sacrificial pile and the pier in different diameter ratios ($D_1/D$). From D1–D5, the diameter ratio was 0.6, 0.8, 1.0, 1.2, and 1.5, respectively. The gravel size was 0.2–0.25 D and the layout distance was 3.0 D. It can be seen from Figure 10 that with the increase of diameter ratio, the area of sediment siltation behind the permeable pile becomes larger and larger, and the local scour depth and range of piers decrease gradually. However, the scour depth and range of permeable sacrificial pile increase gradually.

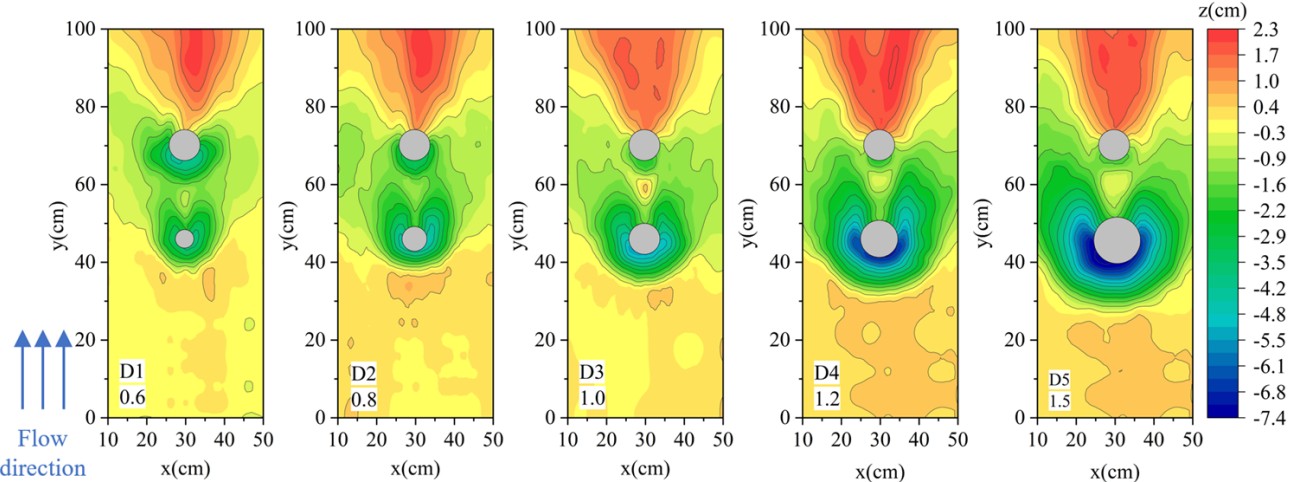

**Figure 10.** Scour around pier protected by permeable sacrificial pile with different diameter ratios.

Figure 11 shows the local scour depths of the sacrificial pile and the pier in different diameter ratios. When the diameter ratio is 0.6, the local scour depth around the pier is the largest while the local scour depth around the permeable sacrificial pile is the smallest. The local scour depth around the permeable sacrificial pile increases linearly with the increase in diameter ratio. Meanwhile, the local scour depth around the pier decreases rapidly at first and then tends to be stable. When the diameter ratio reaches 1.0, the local scour depth around the pier does not change much along with the increase of the diameter ratio.

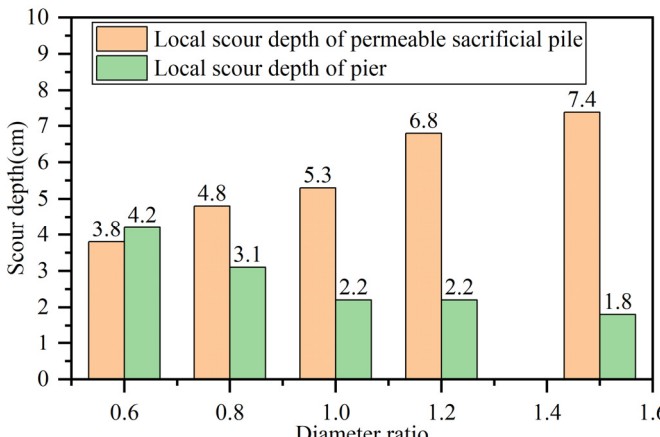

**Figure 11.** Local scour depth around the pier and permeable sacrificial pile with different diameter ratios.

When the diameter ratio increases, the water-blocking area caused by the permeable sacrificial pile increases gradually. This causes increase of downward flow, the local scour depth, and range around the permeable sacrificial pile as well. On the contrary, the protection range of the permeable sacrificial pile is enlarged, and the local scour depth around the pier decreases gradually. If the diameter ratio is greater than 1.0, the protection area still increases. However, the local scour depth of the pier does not change significantly because the local scour area of the pier has already been included in the protection area.

### 3.5. Influence of Submergence Rate on Protect Effect of Permeable Sacrificial Pile

Figure 12 shows the local scour around the pier and the sacrificial pile using permeable sacrificial pile with different submergence rates, in conditions where the gravel size was 0.2–0.25 D, the layout distance was 3.0 D, and the diameter ratio ($D_1/D$) was 1.0. The submergence rate of E1-E5 was 0.2, 0.4, 0.6, 0.8, and 1.0, respectively. As the submergence rate increases, the scour depth and scope around the permeable sacrificial pile increase, however the scour depth and scope around the pier decrease gradually. When the submergence rate reaches to 0.6, the change is no longer significant.

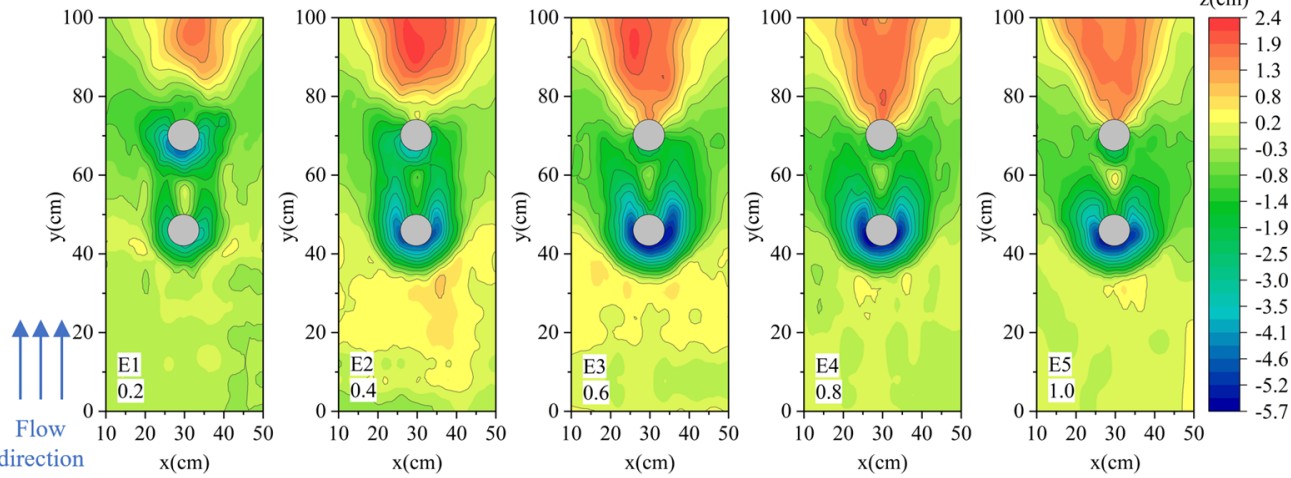

**Figure 12.** Scour around pier protected by permeable sacrificial pile with different submergence rates.

Figure 13 shows the local scour depths of permeable sacrificial pile and pier in different submergence rates conditions. The local scour depth around the permeable sacrificial pile increases when the submergence rate is less than 0.6 and becomes stable when the submergence rate is greater than 0.6. The local scour depth around the pier decreases when the submergence rate is less than 0.6 and tends to be stable when the submergence rate exceeds 0.6.

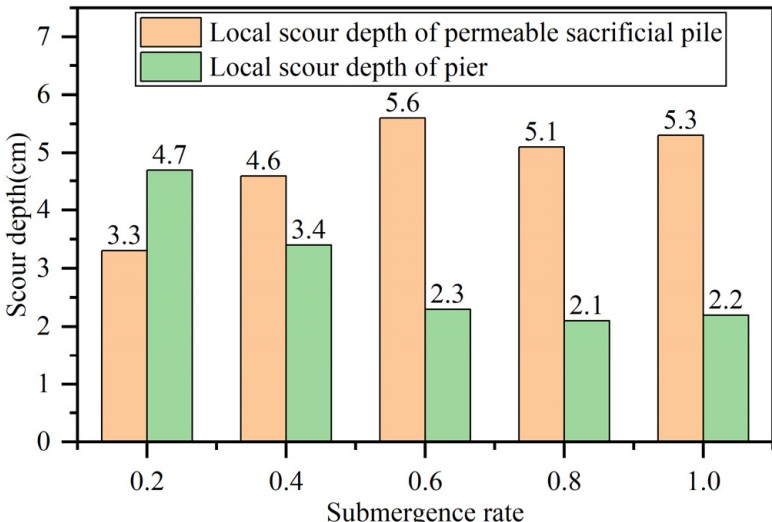

**Figure 13.** Local scour depths around the pier and permeable sacrificial pile with different submergence rates.

When the submergence rate is less than 0.6, the height of the permeable sacrificial pile above the sediment surface is smaller, the water-blocking area is smaller as well. Therefore, there is less downward flow in front of the pile. This explains the reduction of the local scour depth around the permeable sacrificial pile. However, when the height of the permeable sacrificial pile above the sediment surface is smaller, there is more water flow over it. A larger local scour depth is caused around the pier behind. With the increase of submergence rate, the amount of downward flow in front of the permeable sacrificial pile increases gradually, which causes the increase of local scour depth around the permeable sacrificial pile. In other words, as the amount of water flowing over the permeable sacrificial pile decreases, the local scour depth of the pier decreases accordingly.

## 4. Numerical Simulation

### 4.1. Simulation Verification

To understand the mechanism of scour reduction by adopting a permeable sacrificial pile, the flow field characteristics around permeable sacrificial piles and piers were simulated with ANSYS Fluent in fixed bed conditions. It is difficult to make a quantitative comparisons and verifications with existing results due to the different research conditions. So qualitative comparison and verification method were used to prove the rationality and reliability of the numerical simulation.

The simulation in this paper was compared with the results of Melville's model test [24]. Figure 14a,b show both the distribution of the flow field around the pier and that in the front of the pier in Melville's model test and this study. The simulation results are highly consistent with that of Melville's model test. In addition, Roulund [25] pointed out that the flow field around a vertical pile fixed on the bed mainly included the following four flow characteristics: (1) the formation of the bottom boundary layer; (2) the formation of the horseshoe vortex in front of the cylinder; (3) the forming of a new boundary layer in front of the cylinder due to the falling flow; (4) the separation of the flow on the side of the cylinder and the formation of the wake vortex. We can observe the following characteristics from Figure 14: (1) the flow separation on both sides of the pier and a wake vortex is formed within 1 D behind the pier, shown in Figure 14a; (2) the horseshoe vortex in front of the pier and the bottom boundary layer, shown in Figure 14b. This validated the effectiveness and accuracy of the simulation methodology in this study.

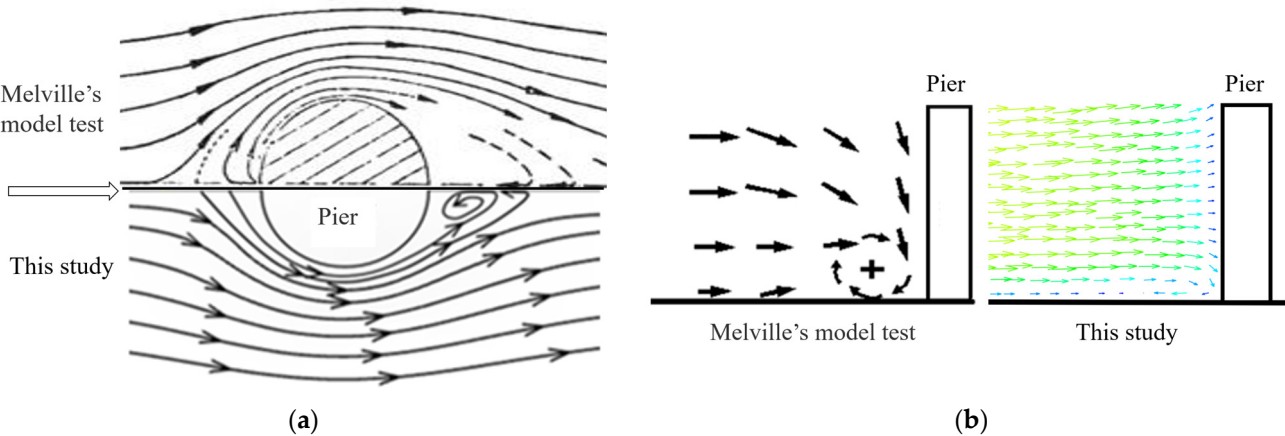

(**a**)                                                                                 (**b**)

**Figure 14.** Comparison of flow field characteristics near the piers. (**a**) Flow field around the pier. (**b**) The flow field in front of the pier.

### 4.2. Model Setup

Experiment A and C2 were simulated in this study. In the simulation to experiment C2, the gravel size was set to 0.2–0.25 D, the distance between the pile and the pier was 3.0 D, the diameter ratio was 1.0, and the submergence rate was 0.6. The model was built via Design Modeler.

Based on Melville's research [24], the width of the domain in the numerical model was set to 10.0 D, which was greater than the requirement of 8.98 D in his. The distance between the entrance and the center of the sacrificial pile was 7.0 D so that the flow could be fully developed. According to the conclusions from Sarker [26], the downstream flow of the cylinder is not affected if the distance behind the pier is greater than 12.0 D. Therefore, the distance from the outlet to the pier center was 15.0 D in this study. The inlet velocity and the water depth of the numerical simulation were the same as in the experiments. The layout and grid division of the numerical model of C2 is shown in Figure 15.

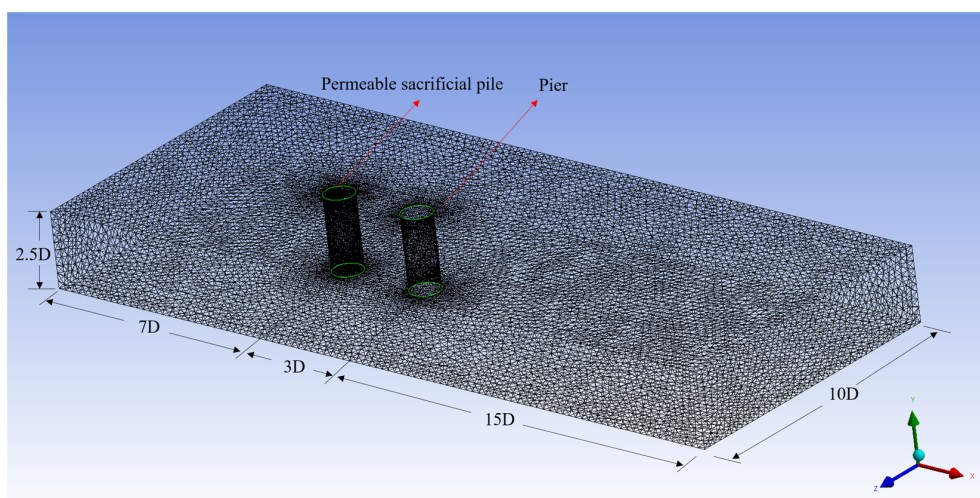

**Figure 15.** Computing domain and meshing of the model of C2.

### 4.3. Boundary Conditions and Input Parameters

The velocity inlet was set as the boundary condition of inlet. The velocity was 0.35 m/s, which was measured 2 cm below the water surface in the experiments. The outlet boundary was set to the pressure outlet. The bottom of the model was a rough wall, and the effective rough height was 2.0 $d_{50}$ ($d_{50}$ is the median particle size of the sediment in the experiment). The sides of the domain were set as the wall boundary. The top surface of the model was set as pressure boundaries. The permeable sacrificial pile was set to porous.

The viscous resistance coefficient and inertial resistance coefficient of porous material can be calculated by the Ergun formula:

$$\frac{1}{\alpha} = \frac{150(1-\varepsilon)^2}{\varepsilon^3\ d_p^2} \tag{1}$$

where: $1/\alpha$ is the coefficient of viscous resistance, $\varepsilon$ is the porosity, and $d_p$ is the equivalent diameter of porous material.

$$C_2 = \frac{3.5(1-\varepsilon)}{\varepsilon^3\ d_p} \tag{2}$$

where: $C_2$ is the inertial resistance coefficient.

The porosity of the permeable sacrificial pile in $C_2$ was 0.491, and the equivalent diameter was 0.018 m, so the corresponding viscous resistance coefficient was 1,013,298.35 and the inertial resistance coefficient was 836.12 in the simulation.

### 4.4. Simulation Results and Analysis

### 4.4.1. Velocity of the Flow Field

The RNG $k-\varepsilon$ model was adopted in the simulation and CFD Post was used for post-processing to obtain the final numerical simulation results. Figure 16 shows the velocity vector and streamlines of the flow field 0.05 m above the riverbed. Figure 16a shows the flow field of experiment A, and Figure 16b shows that of experiment C2. From Figure 16a, the characteristics of the flow field around the solid sacrificial pile and the pier are in good agreement with existing research results, including the lower-speed flow in front of the pile and pier, higher-speed flow on both sides of the sacrificial pile and pier, and lower-speed area and vortex behind them. The shielding of the sacrificial pile plays an important role in reducing the local scour depth of the pier. The higher-speed flow around both sides of the pier is significantly smaller than that on both sides of the solid sacrificial pile because of the shielding effect.

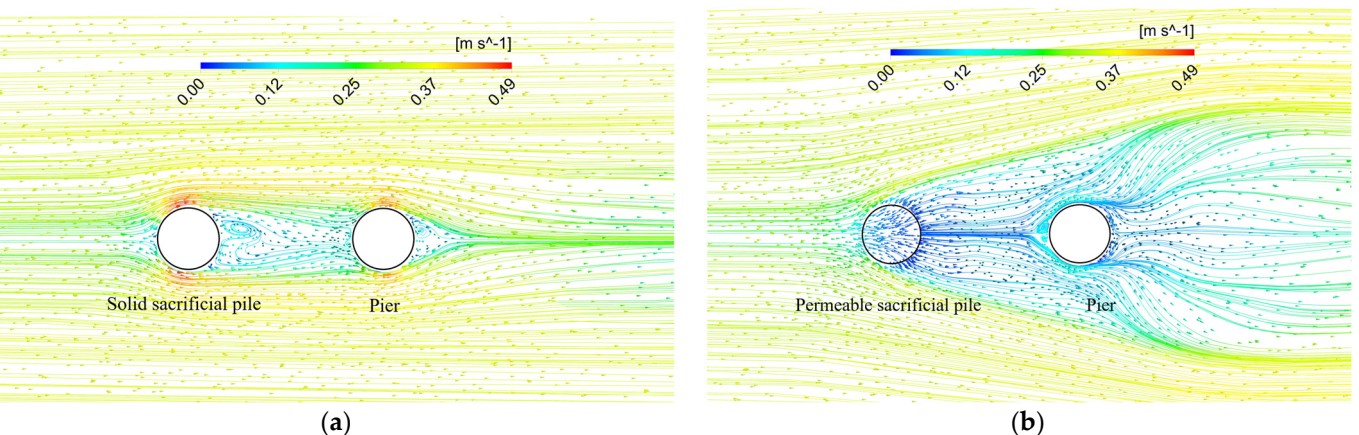

**(a)**　　　　　　　　　　　　　　　　　　　　　　　　**(b)**

**Figure 16.** Flow velocity vector and streamline at z = 0.05 m above the riverbed. (**a**) Experiment A. (**b**) Experiment C2.

Figure 16b shows the flow field of experiment C2. Because of the permeability of the sacrificial piles, part of the water flows through the permeable piles at a lower speed, which reduces the amount and the velocity of downward flow in front of the pile significantly. Therefore, the local scour depth in front of the permeable sacrificial pile decreases. Meanwhile, as shown in Figure 17, because of the difference in the pressure in front of and inside the permeable sacrificial pile, the water flows outward the pile in the direction that is perpendicular to the isobars, which causes a weakness of the flow velocity around the pile, especially in the higher velocity area compared with that in experiment A. Meanwhile, the conflict of water from inside and outside the permeable pile enlarges the low-velocity area around the pile comparing to experiment A, which expands the lower-velocity area

around the pier indirectly. Therefore, the expansion of lower speed area behind the permeable sacrificial pile and around the pier has a significant effect on the scour reduction in the downstream area, especially around the pier. This is in good agreement with the experiments results in Section 3.2.

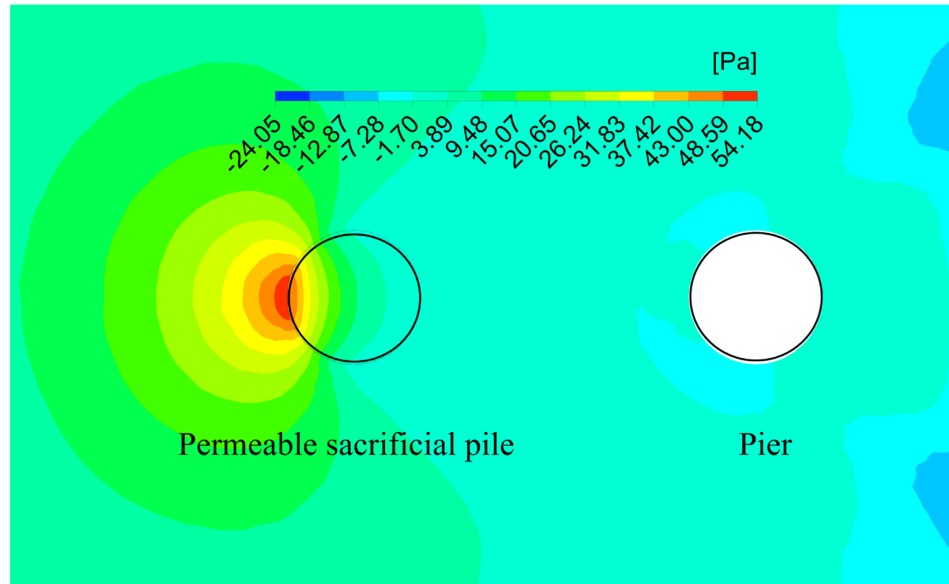

**Figure 17.** Flow pressure at z = 0.05 m above the riverbed in Experiment C2.

Figure 18 shows the velocity vector and streamline at the central longitudinal section of the model. It can be seen that the permeability of the sacrificial pile can not only weaken the downward flow in front of the pile significantly but also has a significant impact on the flow pattern in the region between the sacrificial pile and the pier. Figure 18a shows the velocity vector and streamline of experiment A. A large-scale clockwise vortex was formed near the riverbed and another large-scale counterclockwise vortex was formed near the free surface. The two vortexes separated the water flows to the pier into upward and downward flows. The downward flow caused local scour around the pier. Figure 18b shows that of experiment C2. A small-scale clockwise vortex was formed near the riverbed close to the permeable sacrificial pile, and another large-scale counterclockwise vortex was formed in the middle of the region closer to the pier. The upward flow and the downward flow in front of the pier cancelled each other out, so the local scour between the pile and pier, as well as that around the pier were reduced.

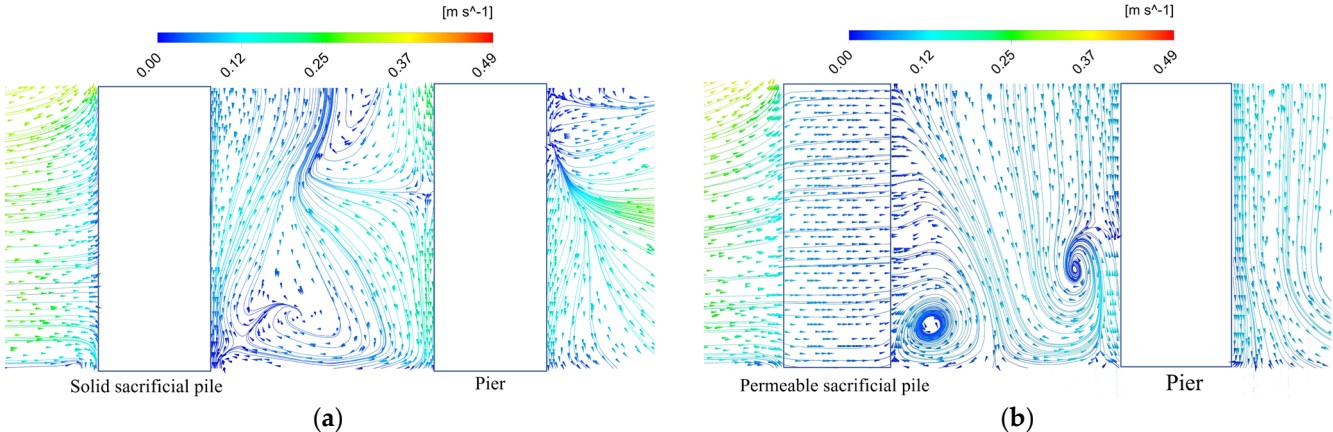

**Figure 18.** Velocity vector and streamline at the central longitudinal section of the model. (**a**) Experiment A. (**b**) Experiment C2.

#### 4.4.2. Distribution of Shear Stress on the Riverbed

Figure 19a,b show the distributions of shear stress near the riverbed in experiments A and C2, respectively. In experiment A, the flow velocity on both sides of the sacrificial pile and the pier increased significantly when water flowed around them, which resulted in two higher-shear stress areas over the riverbed. This explains the local scour around the solid sacrificial pile and the pier. The lower-shear stress area behind the solid sacrificial pile is smaller, and the shape is close to a rectangle with a length of about 1.5 D and a width of about 1.0 D. The lower-shear stress area behind the pier is relatively scattered.

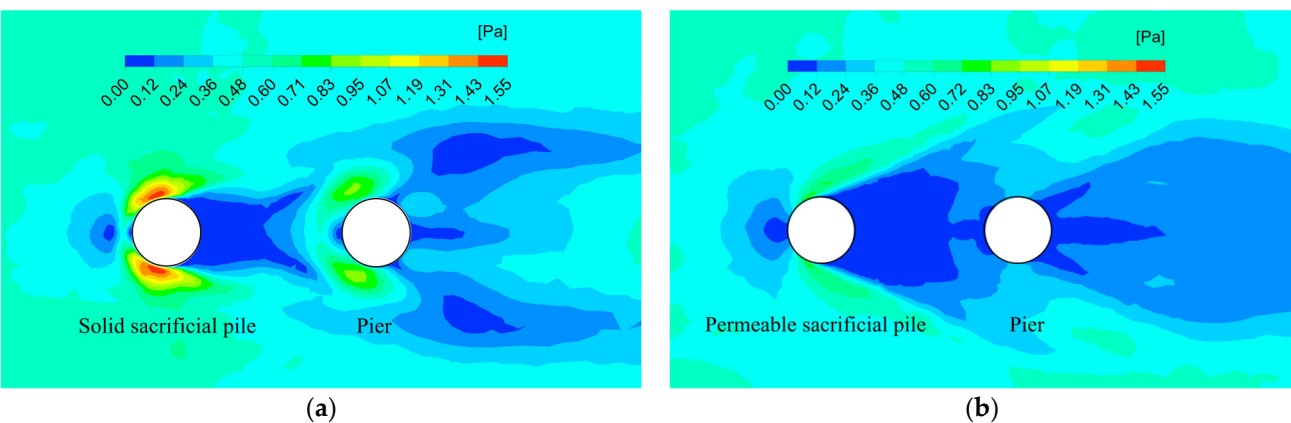

**(a)**                                                                                           **(b)**

**Figure 19.** Shear stress distributions near the riverbed. (**a**) Experiment A. (**b**) Experiment C2.

Compared with the distribution of shear stress on the riverbed in experiment A, there are two main differences in experiment C2. Firstly, there was only one higher-shear stress area on the riverbed on both sides of the sacrificial pile. The maximum shear stress was about half of that in experiment A. The reduction of shear stress is the main reason for the local scour reduction of the permeable sacrificial pile. The maximum shear stress on both sides of the pier was about a quarter of that in experiment A. Secondly, the range of lower-stress areas behind the permeable sacrificial pile was larger, and the shape was close to a triangle with a length of about 2.0 D and a maximum width of 3.0 D. The lower-shear stress area behind the pier was relatively continuous. Above all, the introduction of permeability is the main reason for those differences.

### 5. Conclusions

The influences of the filled gravel size, layout distance, diameter ratio, and submergence rate of the permeable sacrificial pile on local scour reduction in clear water were studied in this paper based on indoor experiments and numerical simulations. The main conclusions are as follows:

(1)  In the same conditions, the reduction effect of the permeable sacrificial pile on the local scour of the pier is close to that using the solid sacrificial pile. However, the local scour depth around the permeable sacrificial pile is smaller than that of the solid sacrificial pile. That is to say, the permeable sacrificial pile has an excellent effect on reducing local scour on itself while providing close protection to the pier

(2)  As the size of the filling gravel of the permeable sacrificial pile increases, the local scour depth around the pier shows a trend of decreasing and then increasing. The local scour depth is the minimum when the size of the gravel is 0.2–0.25 D. The trend of the local scour depth in changing with the distance between the pier and the pile is the same as the trend with gravel size. The local scour depth is the smallest when the distance is 3.0 D. The local scour reduction effect of permeable piles increases with the increase of diameter ratio. The amount of local scour depths around the sacrificial pile itself and the pier are the smallest when the diameter ratio is 1.0, and do not change much when the diameter ratio is greater than 1.0. The local scour reduction effect

increases with the increase in the submergence rate and reaches its best when the submergence rate is 0.8.

(3) Compared with the test case using a solid sacrificial pile, the permeability of the sacrificial pile weakens the downflow in front of the pile and the velocity of the flow around the pile effectively. Thus, the local scour around itself can be reduced. The pressure difference inside and outside the permeable pile leads the water to flow along the direction which is perpendicular to the isobars. This impacts the velocity of the flow, the vortex system, and the shear stress on the riverbed around the pile and pier significantly. The lower-velocity area around the pile expands a lot compared to that around the solid sacrificial pile, which expands the lower-velocity area around the pier indirectly.

(4) Compared with the shear stress on the riverbed in experiment A, the maximum shear stress on both sides of the permeable sacrificial pile is about half of those of the solid sacrificial pile, and the maximum shear stress on both sides of the pier in experiment C2 is about a quarter of that on both sides of the pier in experiment A. Meanwhile, there is a larger lower-stress area behind the permeable sacrificial pile in experiment C2.

(5) Limitations of this study include: the conclusions of the numerical simulation are qualitative and only used to reveal the mechanism of local scour reduction of permeable sacrificial pile. This study is not based on prototypes and cannot be applied to real cases to a certain scale.

**Author Contributions:** H.Q., G.C., W.Z., T.Y., W.T., and J.L. worked together. Conceptualization, H.Q.; Formal analysis, H.Q., W.Z., W.T. and Jiachun Li; Funding acquisition, H.Q. and Jiachun Li; Test and Software, G.C., W.Z., T.Y. All authors have read and agreed to the published version of the manuscript.

**Funding:** This work was supported by the National Key R&D Program of China [Grant Numbers: 2022YFC3002600]; National Natural Science Foundation of China [Grant Numbers: 51708043]; and Natural Science Basic Research Plan in Shaanxi Province of China [Grant Number: 2019JQ-680].

**Institutional Review Board Statement:** Not applicable.

**Informed Consent Statement:** Not applicable.

**Data Availability Statement:** Not applicable.

**Conflicts of Interest:** The authors declare no conflict of interest.

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
