# Peer review of "Characteristics and Mechanism of Local Scour Reduction around Pier Using Permeable Sacrificial Pile in Clear Water"

_water, doi:10.3390/w14244051_

Round 1
Reviewer 1 Report
In this article, the authors studied permeable sacrificial piles filled with stones. The influence of permeable characteristics, the layout distance, the diameter ratio, and the submergence rate of the permeable sacrificial pile on the local scour reduction were studied. The local scour reduction mechanism was revealed based on indoor experiments and numerical simulations. The results can provide a reference for local scour protection of piers.
I suggest that the authors must take into account the following corrections/suggestions:
1. There are some grammatical and typo errors in the paper; the paper should be free from all errors.
2. Polish the abstract by presenting only significant and key outcomes.
3. The author should explain why the study is useful with a clear statement of novelty or originality by providing relevant information in the introduction and conclusion sections.
4. Check figures 4 and 6 again. Is there anything missing?
5. The author should add more discussions on graphs and physical features of the obtained results.
6. Rewrite the conclusion by explaining all the major outcomes of the obtained results on permeable sacrificial piles.
7. A professional proofreading is required for the whole manuscript.
Concluding Remark:
From my point of view, the results presented are new and interesting. Hence, I recommend the paper for publication. However, the above comments or suggestions must be incorporated before the paper's publication.
Reviewer 2 Report
The topic addressed in the manuscript "Characteristics and mechanism of local scour reduction around pier using permeable sacrificial pile in clear water" is very interesting and apparently the results can be very useful for the design of bridge piers. Here are some points that I would like the authors to clarify:
1. How many times was each test performed? Number of replicates
2. I used the information presented by the authors and some information I assumed. I came up with some very strange values, for example, for the characteristics of the river bottom, and I would like the authors to elaborate a little on: scale effects, turbulence behaviour, size of the gravel or rock at the bottom of the river, etc.
3. The authors do not provide the mechanical characteristics of the porous piles: e.g. porosity, permeability, shape factor of the gravels, etc.
4. In some of the tests carried out in the laboratory, water flow values are not given.
5. With the precision of the cases analysed, how representative are the decimals, even tens of units, e.g. in the resistance coefficient of viscosity.
6. In the abstract, the symbol "D" is used without having been defined beforehand.
7. There is no evidence of calibration and validation of the numerical model used. It is important for the authors to keep in mind that all numerical models have limitations and are sometimes very sensitive to small variations of some parameters.
8. I suggest to check that the information presented is sufficient for any other interested person to replicate the tests and/or numerical modelling.
Reviewer 3 Report
Several important results are revealed in the series of experiments. Therefore, the purpose of the numerical simulation in a fixed bed is not clear. The detail explanation for the necessity of numerical modelling is demanded in the sentence. The revision opinion is as follows;
1) Line 59; Please add a simple explanation about “FLOW-3D”
2) Fig.1 Upper Figure ; Which side is upstream?
3) What is the maximum capacity of flow generator?
4) What material do you employ to form the pile and pier?
5) Fig.6; Why is the accretion of sand layer occurred in the downstream of the main pier?
6) Fig.14; You mentioned suddenly about the “Melville’s model test” What is the main reason to adopt the Melville’s test as the target? You carried out the excellent experimental works in the former session. Why do not use such the excellent data as the verification?
Round 2
Reviewer 1 Report
I recommend the paper for publication.
Author Response
Thank you so much for what you have done for our manuscript.And we thank all the reviewers for your valuable and meaningful comments and suggestions, which improved our manuscript greatly.
Reviewer 2 Report
The model used is very sensitive to the parameters with which it is forced. Although the numerical model has been used very successfully in other applications, without proper calibration and validation the uncertainties are high. Considering the above it does not make sense for many parameters to use decimals. In the conclusion section, the limitations should be mentioned so that it is clear to the reader that the manuscript only presents qualitative results.
On the other hand, it should be emphasised in the text that the scale is 1:1 and that the results cannot be scaled to real cases. Also review the units, e.g. porosity, which is normally reported as a dimensionless parameter.
Author Response
"Please see the attachment.

Reviewer 3 Report
The text is well revised.
Author Response

(The authors gave the same response as above.)
